# BOX-TO-BOX TRANSFORMATIONS FOR MODELING JOINT HIERARCHIES

## ABSTRACT

Learning representations of entities and relations in knowledge graphs is an active area of research, with much emphasis placed on choosing the appropriate geometry to capture tree-like structures. Box embeddings (Vilnis et al., 2018; Li et al., 2019; Dasgupta et al., 2020), which represent concepts as $n$-dimensional hyperrectangles, are capable of embedding trees when training on a subset of the transitive closure. In Patel et al. (2020), the authors demonstrate that only the transitive reduction is required, and further extend box embeddings to capture joint hierarchies by augmenting the graph with new nodes. While it is possible to represent joint hierarchies with this method, the parameters for each hierarchy are decoupled, making generalization between hierarchies infeasible. In this work, we introduce a learned box-to-box transformation which respects the geometric structure of box embeddings. We demonstrate that this not only improves the capability of modeling cross-hierarchy compositional edges, but is also capable of generalizing from a subset of the transitive reduction.

## 1 INTRODUCTION

Representation learning for hierarchical relations is crucial in natural language processing because of the hierarchical nature of common knowledge, for example, <Bird IsA Animal> (Athiwaratkun & Wilson, 2018; Vendrov et al., 2016; Vilnis et al., 2018; Nickel & Kiela, 2017). The IsA relation represents meaningful hierarchical relationships between concepts and plays an essential role in generalization for other relations, such as the generalization of <organ PARTOF person> based on <eye PARTOF of person>, and <organ IsA eye>. The fundamental nature of the IsA relation means that it is inherently involved in a large amount of compositional human reasoning involving other relations.

Modeling hierarchies is essentially the problem of modeling a *poset*, or *partially ordered set*. The task of *partial order completion*, a general term to describe tasks which require learning a transitive relation, was introduced in (Vendrov et al., 2016). The authors also introduce a model based on the *reverse product order* on $\mathbb{R}^n$, which essentially models concepts as *infinite cones*. Region-based representations have been effective in representing hierarchical data, as containment between regions is naturally transitive. Vilnis et al. (2018) introduced axis-aligned hyperrectangles (or *boxes*) that are provably more flexible than cones, and demonstrated state-of-the-art performance in multiple tasks.

Thus far, not as much effort has been put into modeling joint hierarchies. Patel et al. (2020) proposed to simultaneously model IsA and HASPART hierarchies from Wordnet (Miller, 1995). To do so, however, they effectively augmented the graph by duplicating the nodes to create a single massive hierarchy. Their model assigns two boxes $B_{\text{IsA}}$ and $B_{\text{HASPART}}$ for each node $n$, which are unrelated, and therefore misses out on a large amount of semantic relatedness between IsA and HASPART.

In this paper we propose a box-to-box transformation which translates and dilates box representations between hierarchies. Our proposed model shares information between the IsA and HASPART hierarchies via this transformation as well as cross-hierarchy containment training objectives. We compare BOX-TRANSFORM MODEL with multiple strong baselines under different settings. We substantially outperform the prior TWO-BOX MODEL while training with only the transitive reduction of both hierarchies and predicting inferred composition edges. As mentioned above, our model's shared learned features should allow for more generalization, and we test this by training on a subset of the transitive reduction, where we find we are able to outperform strong baselines. Finally, we

perform a detailed analysis of the model's capacity to predict compositional edges and transitive closure edges, both from an overfitting and generalization standpoint, identifying subsets where further improvement is needed.

## 2 RELATED WORK

Recent advances in representing one single hierarchy mainly fall in two categories: 1) representing hierarchies in non-Euclidian space (eg. hyperbolic space, due to the curvature's inductive bias to model tree-like structures) 2) using region-based representations instead of vectors for each node in the hierarchy (Erk, 2009). Hyperbolic space has been shown to be efficient in representing hierarchical relations, but also encounters difficulties in training (Nickel & Kiela, 2017; Ganea et al., 2018b; Chamberlain et al., 2017).

Categorization models in psychology often represent a concept as a region (Nosofsky, 1986; Smith et al., 1988; Hampton, 1991). Vilnis & McCallum (2015) and Athiwaratkun & Wilson (2018) use Gaussian distributions to embed each word in the corpus, the latter of which uses thresholded divergences which amount to region representations. Vendrov et al. (2016) and Lai & Hockenmaier (2017) make use of the reverse product order on $\mathbb{R}^n_+$, which effectively results in cone representations. Vilnis et al. (2018) further extend this cone representation to axis-aligned hyperrectangles (or *boxes*), and demonstrate state-of-the-art performance on modeling hierarchies. Various training improvement methods for box embeddings have been proposed (Li et al., 2019; Dasgupta et al., 2020), the most recent of which is termed *GumbelBox* after it's use of a latent noise model where box parameters are represented via Gumbel distributions.

Region representations are also used for tasks which do not require modeling hierarchy. In Vilnis et al. (2018), the authors also model conditional probability distributions using box embeddings. Abboud et al. (2020) and Ren et al. (2020) take a different approach, using boxes for their capacity to contain many vectors to provide slack in the loss function when modeling knowledge base triples or representing logical queries, respectively. Ren et al. (2020) also made use of an action on boxes similar to ours, involving translation and dilation, however our work differs in both the task (i.e. representing logical queries vs. joint hierarchies) and approach, as their model represents entities using vectors and a loss function based on a box-to-vector distance. The inductive bias of hyperbolic space is also exploited to model multiple relations, Ganea et al. (2018a) learn hyperbolic transformations for multiple relations using Poincare embeddings, and show model improvement in low computational resource settings. Patel et al. (2020), which our work is most similar to, represent joint hierarchies using box embeddings. However, they represent each concept with two boxes ignoring the internal semantics of the concepts.

Modeling joint hierarchies shares some similarities with knowledge base completion, however the goals of the two settings are different. When modeling joint hierarchies you are attempting to learn simultaneous transitive relations, and potentially learn relevant compositional edges involving these relations. For knowledge base completion, on the other hand, you may be learning many different relations, and primarily seek to recover edges which were removed rather than inferring new compositional edges. Still, the models which perform knowledge base completion can be applied to this task, as the data can be viewed as knowledge base triples with only 2 relations. There have been multiple works that aim to build better knowledge representation (Bordes et al., 2013; Trouillon et al., 2016; Sun et al., 2019; Balazevic et al., 2019a). Most relevant, Chami et al. (2020); Balazevic et al. (2019b) recently proposed KG embedding methods which embeds entities in the Poincaré ball model of hyperbolic space. These models are intended to capture relational patterns present in multi-relational graphs, with a particular emphasis on hierarchical relations.

## 3 BACKGROUND

### 3.1 BOX LATTICE MODEL

Introduced in Vilnis et al. (2018), a *box lattice model* (or *box model*) is a geometric embedding which captures partial orders and lattice structure using $n$-dimensional hyperrectangles. Formally, we define the set of boxes $\mathcal{B}$ in $\mathbb{R}^n$ as

$$\mathcal{B}(\mathbb{R}^n) = \{[x_1, x^1] \times \cdots \times [x_d, x^d]\}, \tag{1}$$

where $x_i, x^j \in \mathbb{R}$, and we represent all degenerate boxes where $x_i > x^i$ with $\emptyset$. A box model for a set $S$ is a function $\mathrm{Box} : S \to \mathcal{B}(\mathbb{R}^n)$ which captures some desirable properties of the set $S$. As the name implies, the box lattice model is particularly suited to representing partial orders and lattice structures.

**Definition 1** (Poset). A *partially ordered set*, or *poset*, is a set $P$ along with a relation $\preceq$ such that, for each $a, b, c \in P$, we have

1. $a \preceq a$ (reflexivity)

2. if $a \preceq b$ and $b \preceq a$ then $a = b$ (antisymmetry)

3. if $a \preceq b$ and $b \preceq c$ then $a \preceq c$ (transitivity)

**Definition 2** (Lattice). A *lattice* is a poset where each pair of elements have a unique upper bound called the *join*, denoted by $\wedge$, and a unique lower bound called the *meet*, denoted by $\vee$.

The authors note that there are natural geometric operations which form a lattice structure on $\mathcal{B}$:

$$\mathrm{Box}(x) \wedge \mathrm{Box}(y) := \prod_i [\max(x_i, y_i), \min(x^i, y^i)], \tag{2}$$

$$\mathrm{Box}(x) \vee \mathrm{Box}(y) := \prod_i [\min(x_i, y_i), \max(x^i, y^i)], \tag{3}$$

In other words, the *meet* of two boxes is the smallest containing box, and the *join* is the intersection, or $\emptyset$ if the boxes are disjoint. These geometric operations map very neatly to hierarchies, where the meet of two nodes is their closest common ancestor and the join is the closest common descendent (or $\emptyset$ if no such node exists). The ability of this model to capture lattice structure using geometric operations makes it a natural choice to embed hierarchies.

### 3.2 PROBABILISTIC BOX MODEL TRAINING

In Vilnis et al. (2018), the authors also introduced a *probabilistic* interpretation of box embeddings and a learning method which was improved upon in Li et al. (2019) and Dasgupta et al. (2020). By using a probability measure $\mu$ on $\mathbb{R}^d$ (or by constraining the space to $[0, 1]^d$), one can calculate box volumes as $\mu(\mathrm{Box}(X))$. The pullback of this measure yields a probability measure on $S$, and thus the box model can be imbued with valid probabilistic semantics. In particular, since the box space $\mathcal{B}$ is closed under intersection, we can calculate joint probabilities by computing $P(X, Y) = \mu(\mathrm{Box}(X) \wedge \mathrm{Box}(Y))$ and similarly compute conditional probabilities as

$$P(X \mid Y) = \frac{\mu(\mathrm{Box}(X) \wedge \mathrm{Box}(Y))}{\mu(\mathrm{Box}(Y))}. \tag{4}$$

The conversion from a poset or lattice structure to probabilistic semantics is accomplished by assigning conditional probabilities, namely $a \preceq b$ if and only if $P(b \mid a) = 1$. We note that the properties required of the relation $\preceq$ follow as a natural consequence of the axioms for conditional probability. Apart from providing rigor and interpretability, the calibrated probabilistic semantics also inform and facilitate the training procedure for box embeddings, which is accomplished via gradient descent using KL-divergence with respect to the aforementioned probability distribution as a loss function.

As one might expect, care must be taken to handle the case when boxes are disjoint, as there is no gradient. In (Vilnis et al., 2018) the authors made use of the lattice structure to derive a lower bound on the probability, and (Li et al., 2019) introduced an approximation to Gaussian convolution over the boxes which similarly handled the case of disjoint boxes. (Dasgupta et al., 2020) improves this further by taking a random process perspective, ensembling over an entire family of box models. The endpoints of boxes are represented using Gumbel distributions, that is

$$\mathrm{GumbelBox}(X) = \prod_i [X_i, X^i], \quad X_i \sim \mathrm{MaxGumbel}(\mu_i, \beta), \quad X^i \sim \mathrm{MinGumbel}(\mu^i, \beta), \tag{5}$$

where $\mu, \beta$ are the location and scale parameters of the Gumbel distribution respectively. The MaxGumbel distribution is given by

$$f(x; \mu, \beta) = \frac{1}{\beta} \exp(-\frac{x-\mu}{\beta} - e^{-\frac{x-\mu}{\beta}}), \tag{6}$$

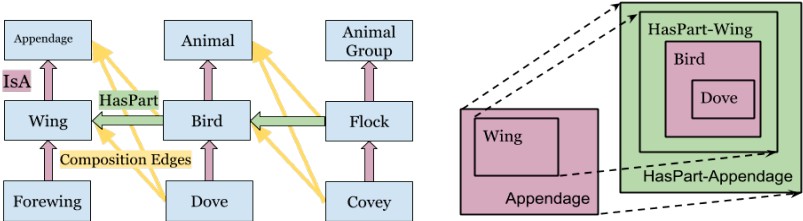

Figure 1: An overview of BOX-TRANSFORM MODEL on joint ISA and HASPART hierarchies. Composition edges are created following certain rules and it should be correctly inferred for a well-trained model. The ISA Wing box is transformed into a HASPART Wing box representing concepts that has wings, and Bird is a subset of it. Same follows for Appendage, and the monotonicity in the ISA space is preserved in HASPART space.

and the $\mathrm{MinGumbel}$ distribution given by negating $x$ an $\mu$. The Gumbel distribution was chosen due to it's min/max stability, making the set of Gumbel boxes closed under intersection, i.e. the intersection of two Gumbel boxes is another Gumbel box. We denote the space of all such boxes as $\mathcal{G}$. The expected volume of a Gumbel box can be efficiently calculated analytically, and in Dasgupta et al. (2020) the authors use this expected volume to calculate the conditional probabilities mentioned in equation equation 4. This training method leads to improved performance on a number of tasks, and is particularly beneficial when embedding trees, thus we will use this Gumbel box approach in our setting.

### 3.3 MODELING JOINT HIERARCHIES

Many existing methods have been proposed for modeling a single hierarchy, however entities are often simultaneously part of multiple hierarchies, for example *hypernymy* (i.e. ISA ) and *meronomy* (i.e. HASPART ). Furthermore, useful information can be shared across inferred compositional edges between the two hierarchies. For example, as shown in 1, based on <Bird,HASPART ,Wing> and <Dove,ISA ,Bird>, we can infer <Dove,HASPART ,Wing>. Due to the compositional nature of these relations, we can infer not only the per-relation transitive closure edges but also the compositional edges, i.e <Dove, HASPART , Wing>.

Formally, for two hierarchical relations $r_1$ and $r_2$, composition edges can be formulated following certain rules. In figure 1, the rules are designed as follows: for <Head,HASPART ,Tail>, $< x_1$, ISA , Head> represent the sub-class of Head, and <Tail, ISA , $x_2 >$ is the super-class of Tail. Composition edges can be generated as $< x_1$,HASPART ,$x_2 >$, $< x_1$,HASPART ,Tail> or $<$ Head ,HASPART ,$x_2 >$. These compositional edges were identified in Patel et al. (2020), where it was observed that a model which effectively captures both hierarchies should make the correct prediction not only over the transitive closure of each individual relation but also on these compositional edges.

## 4 METHODS

### 4.1 BOX-TO-BOX TRANSFORMATION

As mentioned previously, our goal is to not only capture intra-relation transitivity, but also require the model to capture cross-hierarchy compositional edges; that is, for a set $S$ with two partial orders $\preceq_1, \preceq_2$, we want a model capable of learning $(a \preceq_1 b) \wedge (b \preceq_2 c) \implies a \preceq_2 c$ and $(a \preceq_2 b) \wedge (b \preceq_1 c) \implies a \preceq_2 c$ . Furthermore, we hope to do so without including these compositional edges in our training data, in fact we will *remove* parts of these implications in the data, with the expectation that the embedding parameters capture relevant structure which allows us to recover them.

As shown in Dasgupta et al. (2020), Gumbel boxes are able to model hierarchies, we would like to benefit from this capability, particularly for modeling the ISA hierarchy, and thus we seek to learn a

function $f_1 : S \rightarrow \mathcal{G}$, where

$$a \preceq_1 b \iff \frac{E[\mu(f_1(a) \cap f_1(b))]}{E[\mu(f_1(a))]} = 1. \tag{7}$$

For a given Gumbel box,

$$f(x) = \prod_{i=1}^{d}[X_i, X^i], \quad X_i \sim \text{MaxGumbel}(\mu_i, \beta), \quad X^i \sim \text{MinGumbel}(\mu_i + \Delta_i, \beta). \tag{8}$$

where the free parameters are $\mu_i$ and $\Delta_i$. To simultaneously model a second relation, we train a function $\varphi : \mathcal{G} \rightarrow \mathcal{G}$ such that

$$a \preceq_2 b \iff \frac{E[\mu(\varphi(f_1(a)) \cap f_1(b))]}{E[\mu(\varphi(f_1(a)))]} = 1. \tag{9}$$

For notational simplicity, we abbreviate $f_2 = \varphi \circ f_1$.

We choose the transformation $\varphi$ to operate on the "min" coordinate of a Gumbel box and the "side-lengths", that is, we transform a given Gumbel box

$$f(x) = \prod_{i=1}^{d}[X_i, X^i], \quad X_i \sim \text{MaxGumbel}(\mu_i, \beta), \quad X^i \sim \text{MinGumbel}(\mu_i + \Delta_i, \beta). \tag{10}$$

to

$$\varphi\left(\text{GumbelBox}(X)\right) = \prod_{i=1}^{d}[Y_i, Y^i], \tag{11}$$

where

$$Y_i \sim \text{MaxGumbel}(\theta_i \mu_i + b_i, \beta), \quad \text{and} \tag{12}$$
$$Y^i \sim \text{MinGumbel}(\theta_i \mu_i + b_i + \text{softplus}(\theta^i \Delta_i + b^i), \beta), \tag{13}$$

and the $\theta_i, \theta^i, b_i, b^i$ are learned parameters. This effectively translates and dilates the location parameters of the Gumbel distributions which represent the "corners" of a given Gumbel box. We call this model the BOX-TRANSFORM MODEL .

The softplus function is used here as a way to ensure the max coordinate remains larger than the min, and it also provides a simple overflow protection for the expected box volume, as might happen with side-lengths larger than one in high dimensions. While mathematically simple, this transformation allows for parameter sharing between the embedding of a concept with respect to $\preceq_1$ and with respect to $\preceq_2$. Importantly, the transformation is capable of capturing both a global translation and dilation as well as a scaled transformation of the existing learned representation, allowing the absolute position in space (which, for previous box embedding models, was irrelevant) to potentially capture relevant features of the entities.

**Remark 1.** The lack of a transformation on $f_1(b)$ is not an oversight. Using figure 1 as an example, if we consider the Bird box as representative of "all things which are birds", and the HASPART Wing box as the representative of "all thing which have wings", then encouraging containment of the Bird box inside the HASPART Wing box is quite natural. This conceptual motivation is precisely captured by the lack of a transformation on $f_1(b)$. This also coincides with the probabilistic semantics discussed in section 3.2, and is also the method employed by Patel et al. (2020), where this cross-hierarchy containment objective is soley responsible for any flow of information between hierarchies in the TWO-BOX MODEL .

## 4.2 CONNECTION TO TWO-BOX MODEL

There are two main differences between our model and the model introduced in (Patel et al., 2020) which, for reasons which will become clear, we call the TWO-BOX MODEL . First, the TWO-BOX MODEL preceeded the Gumbel box model, and instead uses the Soft box model from (Li et al., 2019). To ensure that the benefits from our model are not conflated with the improvements from using Gumbel boxes we also train a model using the method from (Patel et al., 2020) which makes use of Gumbel boxes.

Table 1: Details of the hypernymy and meronymy hierarchies and the composite edges formed by composition of them.

|  | Transitive Reduction | Transitive Closure | Validation (pos/neg) | Test (pos/neg) |
|---|---|---|---|---|
| Hypernym | 84,363 | 661,127 | 28,838/288,380 | 28,838/288,380 |
| Meronym | 9,678 | 30,333 | 51,64/51,640 | 5,164/51,640 |
| Composite Edge | - | - | 94,807/948,070 | 94,806/948,070 |

Second, both models use different boxes to represent different relations, however the model from (Patel et al., 2020) allows both boxes to have free parameters, relying on containment between boxes representing different relations to pass information. Under the framework we have currently presented, this would be equivalent to learning two functions, $f_1$ and $f_2$, both of which have separate parameters for the min and side length of the boxes for each entity. While such a model has significant representational capacity, we would expect that it would suffer greatly from a lack of generalization. We evaluate this hypothesis by creating a second test, discussed in section 5.4, which removes edges from the transitive reduction of the training data.

## 5 EXPERIMENTS

### 5.1 DATASET

We demonstrate the efficacy of BOX-TRANSFORM MODEL by using the joint hierarchy that has been created by Patel et al. (2020) from WordNet (Miller, 1995). In this dataset, *hypernymy* (ISA ) and *meronymy* (HASPART ) are two hierarchical relations of WordNet over noun sysnets, which are $82, 114$ in total. Individually, the *hypernymy* part of the hierarchy contains $82, 114$ nodes (i.e., all the synsets) with $84, 363$ edges in its transitive reduction and the *meronymy* portion has $11, 235$ synsets (out of $82, 114$ synsets) with $9, 678$ edges in its transitive reduction.

**Joint Hierarchy**   In order to evaluate the performance on the joint hierarchy, Patel et al. (2020) created composition edges using the inter-relational semantics between *hypernymy* and *meronymy*. In particular they use the following composition rules:

$$\underbrace{\text{ISA} \circ \text{ISA} \cdots \text{ISA}}_{\text{0 or 1 or 2 times}} \circ \text{HASPART} \circ \underbrace{\text{ISA} \circ \text{ISA} \cdots \text{ISA}}_{\text{0 or 1 or 2 times}} = \text{HASPART} . \tag{14}$$

To illustrate from Figure 1, *<Dove* ISA *Bird>* ∧ *<Bird* HASPART *Wing>* ∧ *<Wing* ISA *Appendage>* implies that *<birds* HASPART *appendage>*. In total, $189, 613$ composition edges are generated by the method described above for evaluation of the model on the joint hierarchy task. For each test/validation edge, a fixed set of negative samples of size 10 was generated by corrupting the head and tail 5 times each.

We provide the overall statistics for the dataset in the Table 1. We have also created a second training dataset which further removes part of the transitive reduction to evaluate the models on their generalization capability (refer to Section 5.4 & 5.5). The dataset used for those section has different statistics and they are reported in the respective sections.

### 5.2 BASELINE MODELS AND TRAINING DETAILS

We compare BOX-TRANSFORM MODEL against geometric embedding methods as well as knowledge base completion methods. We give a brief description for each baseline below.

1. **TWO-BOX MODEL :** As mentioned in 4.2, Patel et al. (2020) extends the idea of Box embeddings (Vilnis et al., 2018; Li et al., 2019) to model joint hierarchies by defining two boxes per node, one for each relation.
2. **Order Embeddings:** Vendrov et al. (2016) treats each concept as axis parallel cones in positive orthant. We considered two different cone parameters for each entity following the TWO-BOX MODEL (Patel et al. (2020)).

3. **Poincaré Embeddings:** (Nickel & Kiela, 2017) & **Hyperbolic Entailment Cones** (Ganea et al., 2018b): Tree-structured data are best captured in hyperbolic space (Chamberlain et al., 2017). Thus in Nickel & Kiela (2017), the authors learn embedding on $n$-dimensional Poincaré ball. For similar reasons, Ganea et al. (2018b) uses the hyperbolic space however they extend the hyperbolic point embeddings to entailment cones. Again, for these models, two separate parameters are considered for each entity.

4. **TransE and RotatE** (Bordes et al., 2013; Sun et al., 2019): This task can be posed as knowledge base completion for a KB with only two relations. Thus we evaluate TransE and RotatE which are simple yet effective methods for knowledge base embeddings, which achieve state-of-the-art for many knowledge base embedding tasks. Unlike the two box model (Patel et al., 2020) or the other baselines, these methods have shared representation for each entity, and thus they are expected to generalise better on missing edges.

5. **Hyperbolic KG Embeddings** (Balazevic et al., 2019b; Chami et al., 2020): We also compared our method against recently proposed KG embedding methods based on hyperbolic embeddings to model hierarchical structures present in KGs. The Multi-Relational Poincaré model (MuRP) (Balazevic et al., 2019b) learns relation-specific transforms of the entities that are embedded in hyperbolic space. The RoTH (Chami et al., 2020) parameterize the relation specific transformations as hyperbolic rotation, where as the AttH (Chami et al., 2020) combines hyperbolic reflection and rotation using attention. We provide more training related details in Appendix A.1.

## 5.3 COMPOSITION EDGES FROM TRANSITIVE REDUCTION

In order to demonstrate the ability of the model to capture partially ordered (tree-like) data most embedding methods (Ganea et al., 2018b; Nickel & Kiela, 2017; Patel et al., 2020) train their model on the transitive reduction and predict on the transitive closure. For an evaluation on modeling the joint hierarchy, therefore, it is natural to train the models only on the transitive reduction of *hypernymy* and *meronymy* and evaluate on the composition edges, as done in Patel et al. (2020). We report the F1 score (with 1:10 negatives) for those edges in table 2. The threshold used for the classification is determined by maximizing the F1 score on the validation set.

Table 2: Test F1 scores(%)of various methods for predicting the Composition edges.

| Methods | F1 score |
| --- | --- |
| Poincaré Embeddings | 43.8 |
| Hyperbolic Entailment Cones | 44.0 |
| TransE | 57.0 |
| RotatE | 51.0 |
| Order Embeddings | 68.5 |
| MuRP | 21.4 |
| AttH | 51.3 |
| RotE | 51.5 |
| RotH | 55.8 |
| TWO-BOX MODEL (Patel et al., 2020) | 68.1 |
| TWO-BOX MODEL (with GumbelBox) | 73.7 |
| BOX-TRANSFORM MODEL | **82.2** |

Table 3: Test F1 scores(%) of various methods for generalization capability.

| Methods | F1 score |
| --- | --- |
| Poincaré Embeddings | 33.5 |
| Hyperbolic Entailment Cones | 36.0 |
| TransE | 57.0 |
| RotatE | 55.0 |
| Order Embeddings | 54.5 |
| MuRP | 20.1 |
| AttH | 27.0 |
| RotE | 48.8 |
| RotH | 46.7 |
| TWO-BOX MODEL (with GumbelBox) | 58.9 |
| BOX-TRANSFORM MODEL | **63.9** |

From Table 2, we observe that BOX-TRANSFORM MODEL outperforms the other baselines by a significant margin. As mentioned in Patel et al. (2020) and so do we observe that in the next section 5.4 that the Poincaré embeddings and Hyperbolic entailment cones do face difficulty in learning when presented only with transitive reduction edges. However, the hyperbolic KG method Atth  RoTH are able to learn the composite edges to a certain extent. The performance gain of RotH over its euclidean counterpart RotE can be attributed to its inductive bias towards modeling hierarchies. The performance of Box embedding method as proposed by Patel et al. (2020) performs at par order embedding method. However using GumbelBox formulation (Dasgupta et al., 2020), we observe significant performance boost as GumbelBox improves the local identifiability of the parameter space.

Still, the capability of the BOX-TRANSFORM MODEL to benefit from shared cross-hierarchy features allows it to substantially outperform even this improved version of the TWO-BOX MODEL . This is likely due to the fact that the inductive bias provided by the transformation is more in line with the data; the model can benefit from the containments learned as a result of the ISA relation, and learn a HASPART transformation which potentially preserves these containments.

## 5.4 LEARNING FROM INCOMPLETE TRANSITIVE REDUCTION

In Patel et al. (2020), and also in our previous experiment, we already observe that box embedding methods are highly capable of to recovering the transitive closure (in our case, composition edges) given the transitive reduction only. In this experiment, we train with even less of the transitive reduction, moving some of these edges to the test set. Now, reconstruction of the closure and the composition edges require models to generalize over the missing parts of the graph. We train on 9175 *meronymy* edges and 80372 *hypernymy* edges and test/validate on an aggregated pool of 251783 edges. Please refer to the Appendix A.2 for details on dataset creation and statistics.
From Table 3, we observe that BOX-TRANSFORM MODEL outperforms all the baseline methods by a large extent. Although the two box model is performing worse than BOX-TRANSFORM MODEL , it is able to beat other baselines. Out of the two Knowledge base completion methods TransE performs the best and achieves comparative performance to two box model. Although the hyperbolic KG embeddings were able to perform well on the composite edges, their generalization performance is relatively lower than other KG embedding methods. We also observe that the RotE model that was under performing in composite edges, outperforms RotH by some margin in this generalization setting. We select the top three best performing methods for further analysis for each type of edges in the graph.

## 5.5 PERFORMANCE ANALYSIS ON DIFFERENT SPLITS

Training on a subset of the transitive reduction showed that our model could generalize to composition edges even with the absence of essential edges to make such prediction. We further perform evaluation analysis using the same training data with the best-performed model selected by maximizing the f1 score on composition edges. We evaluate the model performance on the transitive closure for each hierarchy (ISA and HASPART ), and the composition edges on the joint hierarchy.

Table 4: Single hierarchy F1 score (%) analysis on ISA and HASPART . The overall dataset is the combination of overfitting, generalization and extended generalization

|  | Type | Overall TC(X) | Overfitting TC(X1) | Generalization X-X1 | Extended Generalization TC(X) - TC(X1) -(X-X1) |
|---|---|---|---|---|---|
| **TransE** |  | 52.9 | 52.1 | 66.5 | 46.0 |
| **Two Box Model** | ISA | 47.8 | 58.9 | 19.9 | 22.9 |
| **BOX-TRANSFORM MODEL** |  | **57.3** | 60.0 | 65.9 | 44.4 |
| **TransE** |  | **59.9** | 63.0 | 56.1 | 48.3 |
| **Two Box Model** | HASPART | 51.6 | 54.8 | 40.8 | 37.8 |
| **BOX-TRANSFORM MODEL** |  | 58.8 | 64.2 | 33.4 | 25.4 |

For each single hierarchy, some edges are removed from the transitive reduction $X$ to create the incomplete transitive reduction training data $X1$. Evaluating the transitive closure of $X$ directly evaluates the model's performance on each hierarchy, denoted as $\text{TC}(X)$. This can be further divided into three categories: dataset that evaluates model ability to capture transitive closure of $X1$, $\text{TC}(X1)$, dataset that evaluates model generalization ability on missing edges $X - X1$, and dataset that evaluates model's extended generalization ability on $\text{TC}(X) - \text{TC}(X1)$.

Table 5: Joint hierarchy F1 score (%) analysis. The overall data is the combination of overfitting and generalization.

| | Overall COMP(X, Y) | Overfitting COMP(X1, Y1) | Generalization COMP(X, Y) - COMP(X1, Y1) |
|---|---|---|---|
| **TransE** | 58.8 | 70.1 | 68.6 |
| **Two Box Model** | 62.5 | 72.7 | 63.6 |
| **BOX-TRANSFORM MODEL** | **69.6** | 86.1 | 70.0 |

Composition edges from the joint hierarchy can be analyzed the same way. $\text{COMP}(X, Y)$ represent all the composition edges in the full wordnet dataset, composed by ISA transitive reduction $X$ and HASPART transitive reduction $Y$. It can be further divided into two categories: data that evaluate model overfitting ability to capture $\text{COMP}(X_1, Y_1)$ where $X_1$ and $Y_1$ is the corresponding training ISA and HASPART data in section 5.4, and data that evaluate model generalization ability on learning logical operations $\text{COMP}(X, Y) - \text{COMP}(X_1, Y_1)$. The detailed statistics on each of these splits are provided in Appendix A.3. The evaluation dataset is created by randomly creating negative examples with the pos: neg ratio 1:10. We select the top 3 best models from section 5.4, then choose the threshold that maximized the F1 score for the validation data of each split and report the test F1. As shown in table 4 and table 5, our model performs the best overall across different dataset splits. BOX-TRANSFORM MODEL performs much better on the full transitive closure of ISA , and all the composition edges. In general, BOX-TRANSFORM MODEL performs much better on transitive closure and composition edges by a large margin in all overfitting settings. TransE does better on predicting removed edges from the transitive reduction (which serves more as an analysis of the model's capability, as it is not a typical evaluation for partial order completion), however we note that our model does surprisingly well on the ISA missing edges, which we attribute to the shared semantics between the hierarchy made possible by this box-to-box transformation.

## 6    CONCLUSION

We proposed a box-to-box transformation which facilitates sharing of learned features across hierarchies. We demonstrate the BOX-TRANSFORM MODEL is capable of excellent performance when predicting compositional edges across a joint hierarchy. Furthermore, the model does an excellent job at modeling the transitive closure of each relation independently. In the future, extending from two relations to modeling multiple relations is essential in order to obtain more generalization from hierarchical ISA edges.

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

# A APPENDIX

## A.1 TRAINING DETAILS

In our experiments, we have kept the number of parameters same across all the methods. We use 5 dimensional box embeddings for the Two Box Model (Patel et al., 2020). Since box embeddings are specified using min and side length in the same dimension. Thus we compare with 10 dimensional order embeddings, Poincaré embeddings, and hyperbolic entailment cones. However, since the above mentioned methods has two different number of parameters for each node, we use 20 dimensional vectors for RotatE, TransE to account for that. Our BOX-TRANSFORM MODEL uses 10 dimension box embeddings for similar reason.

**Hyperparameter range:** We use Bayesian hypermeter optimizer with Hyperband algorithm for all the methods using the web interface Biewald (2020). The hyperparameter ranges are $Gumbel\beta \in [0.001, 3]$, Softplus temperature for box volume $T \in [1, 30]$, $lr \in [0.0005, 1]$, batch size $\in \{8096, 2048, 1024, 512\}$, number of negative samples $\in [2, 30]$ for all the methods. For max margin trainging we searched for the $margin \in [1, 50]$.

The best hyperparameters for our method and a few competitive baselines are provided in appropriate **config** files along with the source code. We will make the code public after the anonymity period.

## A.2 DATASET CREATION STEPS FROM SECTION 5.4

In order to remove edges from the transitive reductions, we iterate through the transitive reduction edges of *meronymy*. With 0.5 probability we choose the edge for further processing. For each chosen HASPART edge, we select an outgoing ISA edge and pair them. We drop the ISA edge from the pair with 0.9 probability (the ratio of HASPART to ISA transitive reduction) and drop the HASPART edge in case the ISA is not dropped already.

This procedure ensures that all the edge removals happen around the composition edges, thus, the results reflect the models true capacity to generalize well for this joint hierarchy task. We evaluate the model on the composition edges, the removed reduction edges, and the closure edges with 251783 in numbers which we split into two parts for validation and test. In Table 3, we report the F1 score on this aggregated evaluation data with 1:10 fixed true negatives.

## A.3 DETAILS OF THE SPLITS FROM SECTION 5.5

Table 6: Dataset statistics for different parts of individual ISA and PARTOF hierarchy.

| Hierarchy | TC(X) | TC(X1) | X-X1 | TC(X) - TC(X1) - (X-X1) |
|---|---|---|---|---|
| IsA | 61,667 | 51,195 | 3,991 | 6,481 |
| HasPart | 30,335 | 26,388 | 503 | 3,444 |

Table 7: Dataset statistics for different composition edges in Joint Hierarchy.

| Hierarchy | Comp(X, Y) | COMP(X1, Y1) | COMP(X, Y1) - COMP(X1, Y1) |
|---|---|---|---|
| Joint Hierarchy | 189,613 | 146,867 | 42,746 |

## A.4 VISUALIZATION

We plot 2-dimensional box embeddings to inspect the quality of our proposed BOX-TRANSFORM MODEL . We use the box embedding parameters of the best performing model of experiment 5.3 (Table 2). Note that, the model is 10 dimensional. However, for a perfectly trained model, we should

observe containment along each dimension. Thus, we pick two dimensions randomly out of the 10-d to visualize the embeddings.

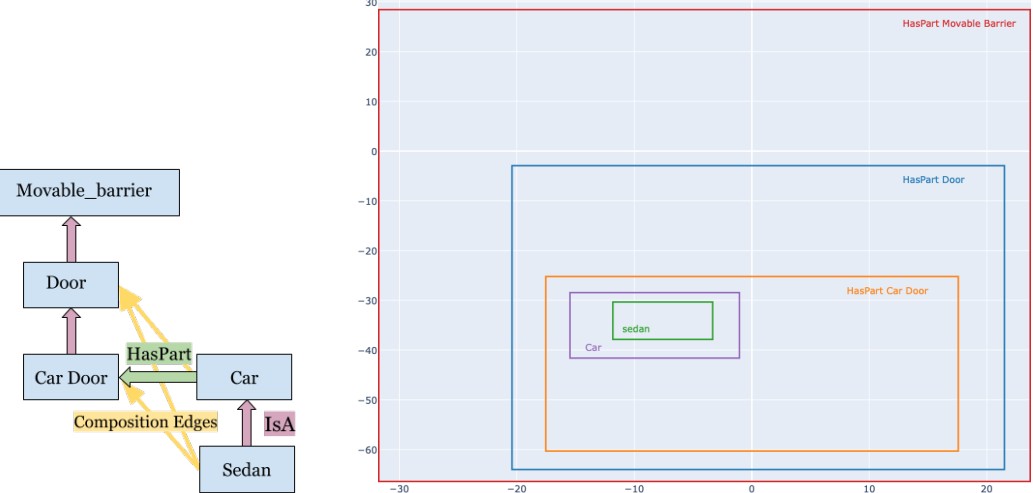

(a) Example of Joint Hierarchy extracted from the WordNet dataset.

(b) We plot the transformed IsA box for "Sedan" & "Car" and transformed HASPART box for "Door", "Car Door", "Movable Barrier" on the same space. The transformations do preserve the containment and provide an consistent assignment of box embedddings for the example on left.

Figure 2: 2-dimensional visualization of proposed Box embedding model.

From the example, the facts that <Car,HASPART ,CarDoor> and <CarDoor,IsA ,Door> would infer that <Car,HASPART , Door>. We observe from the Figure 2 that the HASPART transformation of the "Car Door" and "Door" successfully encloses the IsA transformation of the "Car", thus our model is able infer that composition edge . All the other composite edges such as <Sedan,HASPART , CarDoor>, <Sedan,HASPART , Door> etc. can be similarly inferred from the visualization.

