# OpenReview forum: "Box-To-Box Transformation for Modeling Joint Hierarchies"
_ICLR.cc/2021/Conference — Reject_

### Official Review · AnonReviewer2 · 2020-10-13
**Promising paper with lack of experimental support**

**Rating:** 4
**Confidence:** 4

**Review:**

This work proposes to model multiple hierarchical relations using box embeddings, motivated by the natural transitivity property of the containment between regions in region-based representations. The proposed model is evaluated on a dataset containing two relations (is-a and has-part). Although the proposed model shows promise by outperforming several baselines on the above mentioned dataset, I believe that the paper is not ready for publication in its current form, mainly due to (i) missing comparison to the highly relevant line of work on hyperbolic embeddings of hierarchical multi-relational data; and (ii) lacking additional experiments on a dataset with more than 2 relations.


Detailed comments and questions for the authors:

Sec. 2
1. I find the claim that "Modeling joint hierarchies is not quite the same as knowledge base completion." to be unsubstantiated. This is true to an extent, since the ultimate goal of KB completion is inferring which other facts are true based on existing ones. However, to achieve this goal, KG completion models need to learn entity and relation representations which capture various properties of entities (e.g. semantics) and relations (e.g. transitivity, symmetry, etc.), which is very similar to the main idea of this work.
2. A whole line of very relevant work on hyperbolic embeddings of hierarchical relations in knowledge bases is missing [1, 2].

Sec. 3\
If I understood Definition 2 correctly, the meet (i.e. union) of two boxes will be another box which in most cases contains an area that is not part of either boxes (since a union of two boxes is not necessarily a box). Doesn't this introduce errors into box embeddings which increase with increasing the dimensionality of the embeddings?

Sec. 4\
It is not clear to me why the lack of a transformation on f_1(b) encourages the containment in figure 1. Could you please explain this point further?

Sec. 5
1. While the achieved results seem impressive, as mentioned above, a highly relevant comparison to [1] and [2] is missing. Both missing works are embedding models that represent multiple simultaneous hierarchies in hyperbolic space and where entity embeddings are shared across relations, which should lead to better generalisation on missing edges (as claimed by the authors).
2. The authors evaluate the proposed model on a single dataset with only 2 relations. The proposed model should be evaluated on at least one more dataset, e.g. WN18RR [3], since [1] show several relations in that dataset to be hierarchical.
3. I'm surprised that the improvement over the TWO-BOX model is lower when testing for generalisation capability (5%) than in the original setup (8.5%), given the original premise that the proposed model should benefit from sharing information across hierarchies.
4. It would be nice to see a visualisation of the learned embeddings.

Minor comments:\
Background section should be made shorter, especially the part regarding the probabilistic box model training, which is not that relevant to the overall goal of this work. This space could be used for additional experiments proposed above.

[1] Balazevic et al. Multi-relational Poincaré Graph Embeddings, NeurIPS 2019\
[2] Chami et al. Low-Dimensional Hyperbolic Knowledge Graph Embeddings, ACL 2020\
[3] Dettmers et al. Convolutional 2D Knowledge Graph Embeddings, AAAI 2018

---

> ### Author Response · Authors · 2020-11-18
> **Hyperbolic graph embeddings evaluated for modeling joint hierarchies**
>
> Thank you for your detailed comments, we will address them invidividually below.
>
> Sec 2:
> 1. We agree that KB completion often requires modeling hierarchical relations, and moreover we evaluate KB completion models as our baselines for this task. Our intent was not to say they are entirely unrelated, but rather to simply point out the differences, which, as you point out, are mostly in the amount of emphasis on hierarchical relations and modeling goals. In short, we believe we are essentially in agreement on this point, and have attempted to clarify the wording of this in the paper.
> 2. Thank you for suggesting the hyperbolic embeddings for multi relational knowledge bases. We have added them to the related work section, and also evaluated them as baselines. (Also discussed again below.)
>
> Sec 3:
> As you point out, the meet is not actually the union, it is the smallest containing box, however (1) the meet operation serves primarily to justify the theoretical properties of the box as a lattice, (2) we do not directly train or evaluate using the meet operation, as it is not needed for our queries. That being said, for a well-trained model of hierarchies, the meet is still meaningful, as the meet of a node and one of it’s descendents would simply be the node itself, and the meet of any two arbitrary nodes will provide the smallest containing box which, itself, is contained in the closest common ancestor node.
>
> Sec 4:
> We have updated the image and explanation. The lack of a transformation on f_1(b) means that the transitive relations present in the IsA hierarchy interact exactly as desired with the HasPart hierarchy to encourage the composition edges. For example, since Dove is contained in Bird, if Bird is contained in HasPart-Wing, then Dove will also be contained in HasPart-Wing. A transformation on f_1(b) would add flexibility to the model but would no longer guarantee these compositional edges. Furthermore, if we think of the Bird box as representative of “all things which are birds” and the HasPart-Wing box as “all things which have wings”, then we don’t want or need an additional transformation on the Bird box in this scenario, as “all things which are birds” is a subset of “all things which have wings”.
>
> Sec 5:
> 1. We have run additional experiments using MuRP[1], RotH, and AttH[2], and updated Table 2 and 3 with the results. We observe that the RotH and AttH models were able to learn the joint hierarchy to some extent, however, their generalisation performance is poor. We are also running RotE (the euclidean embedding version of the RotH) to investigate how much the inductive bias of the hyperbolic embeddings is helping in this task.
> 2. The hierarchical relationships in WN18RR are member_meronym, has_part, instance_hypernym, and hypernym, which are already present in our dataset (member_meronym and has_part are both HasPart, instance_hypernym and hypernym are IsA). The other dominant relationships are _derivationally_related_form, _verb_group, and _similar_to, which are symmetric in nature. Furthermore, more than 90% of the evaluation data coming from these relations have a reverse edge present in the training data, which makes modeling these relations trivial [1]. Most of the models achieve 0.93 MRR performance on this subset, including our method that has no inducative bias towards modelling symmetry.
> Removing this trivial symmetric subset and focusing exclusively on hierarchical data in the training split WN18RR yields a large number of connected components, as opposed to a deep hierarchy, and thus is not suitable to our goals of assessing the ability of a model to handle hierarchical relations.
> [1] Pezeshkpour et.at. Revisiting Evaluation of Knowledge Base Completion Models, AKBC 2020.
> 3. In the overfitting task, we train on the whole transitive reduction and predict the performance of the composite edges. However, in the generalisation task we provide a subset of the hierarchies as training data and try to predict on the composition edges and missing edges as well. Thus these numbers are not a direct indication of the generalization gap since the training data is different for these two settings. For this reason, we study the generalization performance for different parts of the dataset in detail in section 5.5.
> 4. We are generating a visualization of the learned embeddings and will include it in the paper shortly.

---

> > ### Author Response · Authors · 2020-11-19
> > **Visualization**
> >
> > We have added the visualization of the learned embeddings and transformations for an example extracted from the wordnet dataset. This visualization has been added to Appendix A.4 of the revised version.

---

> > > ### Comment · AnonReviewer2 · 2020-11-19
> > > **Response to authors after rebuttal**
> > >
> > > Thank you for taking the time to respond to my questions and updating the paper. I have 2 additional questions/concerns:
> > >
> > > 1.  What is the dimensionality used for the multi-relational hyperbolic models compared to box embeddings? I'm surprised by their relatively low performance, which is in some cases even worse than that for single-relational hyperbolic models, which is unexpected.
> > >
> > > 2. Even though the dataset used by the authors contains some of the relations present in WN18RR, that does not change the fact that the dataset used contains only 2 relations. Additionally, given that the subtypes member_meronym and has_part are both represented by the HasPart relation and the subtypes instance_hypernym and hypernym by IsA, it is impossible to conclude whether the proposed model would be able to differentiate between those subtypes (e.g.  member_meronym and has_part) if they were separated. Reporting results on WN18RR (or another hierarchical multi-relational dataset) would help demonstrate: (i) whether the proposed model is capable of modelling more than 2 hierarchies simultaneously; (ii) whether the performance of the model degrades when some of the relations aren't hierarhical, which is important given that is the case with the majority of real-world KG datasets; and (iii) how the proposed model compares to the hyperbolic multi-relational models on a well-established KG benchmark.
> > >
> > > For now, I am reluctant to change my original rating, but I would be happy to do so if the above concerns are adequately adressed.

---

> > > > ### Author Response · Authors · 2020-11-23
> > > > **RE: Response to authors after rebuttal**
> > > >
> > > > Thank you for responding to our reply. We address the concerns individually below.
> > > >
> > > > ### Response to Q.1:
> > > > **Hyperbolic Baseline Parameter Tuning:** Our hyperbolic model used 20-dimensional embeddings to provide a fair comparison (the number of parameters per entity is 20 for all models).  We tuned the other hyperparameters using Bayesian hyperparameter optimization with Hyperband early stopping over the following ranges:
> > > >
> > > > learning rate: [1e-1, 1e-7],
> > > > regularization weight: [1, 1e-7],
> > > > batch size: [256, 512, 1024, 2096],
> > > > negative samples: [2, 100],
> > > > add bias to score: [True, False].
> > > >
> > > > We have added the code (which is directly based on https://github.com/HazyResearch/KGEmb) to the supplementary zip file.
> > > >
> > > > **Why multi-relational hyperbolic does not perform as good as single-relation hyperbolic:**
> > > >
> > > > As observed in Patel 2020, Poincare embeddings require more depth information in order to model the transitive closure of a tree. The compositional edges across two relations in our model can be viewed as the transitive closure of an augmented graph. We hypothesize that the multi-relational hyperbolic embeddings may allow more flexibility in the sort of representable graph structure across different relations. While this is more desirable for other tasks (eg. KBC), it may mean that the model is even less biased toward modeling hierarchies that interact in this way.
> > > >
> > > > ### Response to Q2.1:
> > > >
> > > > We combine various subtypes for two reasons.
> > > > Some are not present in sufficient quantities to evaluate independently;
> > > > Hypernym: 74401 & Instance_Hypernym: 8645
> > > > Part meronym: 10192 & substance_meronym: 1173.
> > > > On their own, instance_hypernym and substance_meronym are not hierarchical. They are simply forests with several connected components with an average max path length of directed graphs formed by them are just over 1. On the other hand, Hypernym and Part_meronym are indeed hierarchical with max path length of 19 and 5 respectively. However, combining these insufficient relations with their dominating counterparts strengthens the overall hierarchical nature of the graph, e.g., combining instance_hypernym helps hypernym to become one single hierarchy with only 1 connected component.

---

> > > > > ### Author Response · Authors · 2020-11-23
> > > > > **RE: Response to authors after rebuttal (cont.)**
> > > > >
> > > > > ### Response to Q2.2:
> > > > > **In general:** The goal of this paper is to jointly model multiple hierarchical relations of sufficient depth and connectedness where there are interesting cross-relational interactions. In particular, while models that perform KBC can also be applied to this task, this is a different task than KBC, where, as we note below, the hierarchical relations are often severely disconnected.
> > > > >
> > > > > More importantly, we do not claim that our model can perform KBC more generally and certainly expect existing KB embedding models to perform better on such tasks.
> > > > >
> > > > > **Performance on WN18RR:**  Although it is not our focus, as requested we have evaluated our model on WN18RR and found that it achieves 37.5 MRR. While not SOTA performance, we find the performance acceptable, given the following:
> > > > > 1. The goal of our model is to model hierarchical relations jointly, but the KBC datasets have many symmetric relations. For example, based on our analysis, over 37.3% of the relations in the WN18RR test set are symmetric.
> > > > > 2. Moreover, none of the relations in WN18RR are, in fact, hierarchical. In [1,2] the Krackhardt hierarchy score for each relation is reported, however the name of this metric is a bit misleading, as the calculation of this metric (Appendix D in [1]) only represents the relation's asymmetry. For example, any graph where each node has at most one edge connecting it to a single other node (i.e. max path length of 1) would have a Krackhardt hierarchy score of 1, despite being a disconnected set of edges. Krackhardt actually proposed four metrics (“hierarchy”, “connectedness”, “efficiency”, and “lubness”), all of which should be 1 for a hierarchical graph [3]. Based on our analysis, the hierarchical relations in all of WN18RR are comprised of a large number of shallow trees.
> > > > > 3. Finally, the split of WN18RR was created with the goal of KBC in mind, and thus the hierarchies in the training data are even further disconnected - it is essentially a forest with many small trees. On the other hand, our dataset split (details of which are in Appendix A.2) removes edges from the training set which would be difficult to reconstruct by chance while still preserving the connectivity of the combined graph with respect to these hierarchical relations.
> > > > >
> > > > > We will include the Krackhardt metrics for our proposed split and how it compares to WN18RR in the camera-ready of this paper.
> > > > >
> > > > > **Modeling more than two relations:** As evidenced by our evaluation on WN18RR, our approach can easily be extended to more than two relations by learning additional transformations. We are currently performing experiments with member_meronym as a third hierarchy and will include results for our method as well as the baselines (including hyperbolic embeddings) in the camera-ready.
> > > > >
> > > > > [1] Balazevic, Ivana, Carl Allen, and Timothy Hospedales. "Multi-relational Poincaré graph embeddings." In Advances in Neural Information Processing Systems, pp. 4463-4473. 2019.
> > > > >
> > > > > [2] Chami, Ines, A. Wolf, D. Juan, F. Sala, S. Ravi and Christopher Ré. “Low-Dimensional Hyperbolic Knowledge Graph Embeddings.” ACL (2020).
> > > > >
> > > > > [3] Krackhardt, David. "Graph theoretical dimensions of informal organizations." Computational organization theory 89, no. 112 (1994): 123-140.

---

> > > > > > ### Comment · AnonReviewer2 · 2020-11-24
> > > > > > **Response to authors**
> > > > > >
> > > > > > Thank you for responding to all the questions and performing additional experiments.
> > > > > >
> > > > > > However, an MRR score of 37.5 on WN18RR is much lower compared to the hyperbolic models (MuRP 48.1 and RotH 49.6) or even those models not specifically designed for modelling hierarchy, which leaves me unconvinced about the ability of the proposed approach to model multiple hierarchies or more than 2 relations in general. Further, the authors argue that "none of the relations in WN18RR are, in fact, hierarchical" which is factually incorrect, especially given that they use 4 of the relations from WN18RR in their own dataset. The WN18RR data may contain missing edges, but the relations themselves **are** hierarchical, and an embedding model designed specifically for modelling hierarchies should be capable of capturing that. I accept that the proposed model may underperform on symmetric relations from WN18RR, but I would expect it to at the very least be able to capture the hierarchical ones, which this paper unfortunately does not show.
> > > > > >
> > > > > > Thus, I stand by my initial score.

---

> > > > > > > ### Author Response · Authors · 2020-11-24
> > > > > > > **Lack of Hierarchical Relations**
> > > > > > >
> > > > > > > There is a distinction to be made between a relation being semantically hierarchical and a set of edges which represent a hieararchy. While hypernymy, for example, is semantically hierarchical, that does not mean that any subset of hypernymy edges would exhibit hierarchical structure from a graph-theoretic perspective, nor that a model which is capable of modeling hierarchies should be capable of representing a given subset.
> > > > > > >
> > > > > > > For example, consider taking a binary tree on N nodes, which is unquestionably hierarchical, and then take any subset of edges $S$ such that each node has out_degree + in_degree <= 1. Further, take a random subset of these edges as a training set $T$, and then evaluate on the remaining edges in $S\setminus T$. There is no reason that a model which is designed specifically for modeling hierarchies (nor, indeed, *any* model) should be capable of recovering the edges in $S$, which is simply a collection of nodes which are each connected to at most one other node. The fact that they were originally from a tree is completely lost, and, probabilistically, *any* extension to a tree which contains the edges in $T$ is just as likely as any other.  The probability for any given element to be the root, for example, is essentially uniform across all nodes with in_degree = 0.
> > > > > > >
> > > > > > > This is an (only slightly) exaggerated version of the situation in WN18RR.

---

> > > > > > > > ### Author Response · Authors · 2020-11-24
> > > > > > > > **Three Hierarchical Relations:**
> > > > > > > >
> > > > > > > > As requested, we have also created an evaluation involving 3 hierarchical relations:
> > > > > > > > IsA (hypernym + instance_hypernym)
> > > > > > > > PartOf (part_meronym + substance_meronym)
> > > > > > > > member_meronym
> > > > > > > > We include compositional edges between member_meronym and IsA in the same manner as for PartOf and IsA. Preliminary experiments on a representation task suggest a similar trend for model performance, where the box-to-box transformation model outperforms multi-relational hyperbolic baselines.
> > > > > > > >
> > > > > > > >
> > > > > > > > | 3-Relation Hierarchy  |   ---------------------------------- --------------| F1 score   |
> > > > > > > >
> > > > > > > >     | MuRP |                                   |   0.17  |
> > > > > > > >     | RotE   |                                   |   0.53  |
> > > > > > > >     | AttH   |                                   |   0.53  |
> > > > > > > >     |  RotH  |                                  |   0.54  |
> > > > > > > >     | Box-to-box transform |                    | 0.71     |
> > > > > > > >
> > > > > > > >
> > > > > > > > If accepted, we will include these results in the camera-ready, along with evaluations on the other baselines as well as generalization evaluations for this three-hierarchy setting.

---

### Official Review · AnonReviewer3 · 2020-10-16
**nice experimental results, lack motivation and training details**

**Rating:** 4
**Confidence:** 4

**Review:**

The paper focuses on modeling multiple hierarchical relations on a heterogenous graph.  The task “modeling joint hierarchies” is essentially trying to infer whether a given pair of entities has a hierarchical connection especially when there exists multiple hierarchical relations (2 in the paper), and missing links. The paper proposes to embed entities using boxes whose endpoints follow the Gumbel distribution. Given there exists two hierarchical relations, the paper transforms the box of one entity under relation 1 to the box of the entity under relation 2 with a parameterized linear function. This is in contrast to previous work that parameterized the box of two relations using separate independent parameters.

The model seems sound, however I have two major concerns. (1) I do not think the model is motivated well, especially on why the model uses Gumbel distribution to parameterize the box. (2) The paper has no introduction how they train the model and use it for inference, what is the loss? This makes it hard to evaluate the correctness of the model.

I am very satisfied with the extensive experiments the paper has conducted. They include many strong baselines including the order embeddings, hyperbolic embeddings and even some KG embeddings. The results on the KG embeddings clearly show that their methods work much better in this (a little specific) hierarchical relation modeling setting. The paper also introduces a new missing-edge setting, where they show that joint modeling achieves better generalization than independent parameters.

Some detailed questions are listed below.
1. The related work states the difference between modeling hierarchies and knowledge base completion, however, it lacks discussion how their Gumbel box is different from previous box embedding methods (this should be added in the second paragraph). I understand the difference between the Gumbel box and the Two-box model, namely the Two-box model learns independent parameters. However, I did not find the discussion on the connection between the Gumbel box and hard/smooth box. Why cannot we apply the same transformation idea to previous hard and smoothed box embeddings so that they can also model joint hierarchies without optimization issues? Why is Gumbel distribution special and useful in parameterizing the boxes and modeling hierarchies?
2. The paper has some vague sentences like “the authors demonstrate that this method of training boxes leads to better representation of trees thus we will use this Gumbel box approach in our setting.” and “since gumbel boxes effectively model hierarchies, we would like to benefit from the inductive bias of this model for intra-relation edges and thus we seek to learn a function ...”, but what is the inductive bias of Gumbel? It’s better to clearly state it.
3. The paper lacks a short discussion and introduction to the Gumbel distribution in the background section, especially on the parameters \mu and \beta.
4. As defined in Eq. 3, the meet of two boxes may include some blank space that does not belong to the input boxes, do you think this will have any issues, especially when the two input boxes are far away from each other?
5. Sec 4.1, first paragraph, “$(a \leq_1 b) \wedge (b \leq_2 c) \to (a \leq_2 b)$” is wrong. Bird has part Wing, and Wing is an Appendage, but Bird is not a Wing.
6. Sec 4.1, end of page 4, “To simultaneously model a second relation, we ...”, so the model can only model two hierarchical relations? If so, I think it is a little limited and can the model provide a way to generalize beyond two hierarchical relations?
7. Sec 4.1, “the free parameters are $\mu_i$ and $\Delta_i$”, why does the model not learn $\beta$?
8. As in Eq. 11 and 12, the transformation is a rather simple linear transformation, have you tried something that is more complex, e.g. a MLP?
9. I am also confused by Remark 1 and Eq. 8. For Bird, there should be two boxes where one represents the IsA relation and the other represents the HasPart relation, right? Then in Figure 1,  why is the IsA-Bird box inside the HasPart-Wing box, I think it should be the HasPart-Bird box inside the HasPart-Wing box.
10. The paper does not introduce how to train the model or even how to make predictions during inference in Sec 4. I understand the page limit but these two aspects are essential to a machine learning model.
11. What is the difference between the two-box model and the order embeddings in the experiments? I assume if you apply the order embeddings to this multi-hierarchical relation setup, then it is the same as the two-box model?
12. I am curious about the performance of the proposed model in an imbalanced dataset (as introduced in Li et al. ICLR 2019), where the ratio of positive and negative is 1:10?

minors: The paper does not have grammar mistakes and here are some minor points.
1. Make it explicit in the introduction that the “Two-Box Model” is referred to Patel et al. (2020)
2. The definition of box lattice model is not self-contained in Eq. 1, what is $x_i$ and $x^i$? I guess it is the two end points of the box in one dimension. Better to state it clearly.
3.  Sec 3.3, “For example, as shown in 1, based on…” -> “For example, as shown in Figure 1, ..”

---

> ### Author Response · Authors · 2020-11-18
> **Clarification regarding Gumbel Box**
>
> Thank you very much for your detailed comments, we have addressed them individually below.
>
> 1. GumbelBox was introduced by Dasgupta et al., 2020, where the authors demonstrate that it solves problems related to local non-identifiability and smooths the loss landscape of prior methods (hard and smooth boxes). In that work, the authors choose a Gumbel distribution because it is min/max stable, and therefore boxes whose min/max endpoints are parametrized via Gumbel distributions are closed under intersection (i.e. the intersection of two Gumbel boxes is another Gumbel box). We have clarified these points in the background section. All of our experiments are carried out using GumbelBox, as Dasgupta et al. 2020 shows it outperforms HardBox and SmoothBox in all tasks. As you rightly point out, the same transformation can be applied to SmoothBox and HardBox, in fact Dasgupta et al. 2020 point out that SmoothBox and HardBox can be viewed as special cases of GumbelBox for specific settings of hyperparameters (i.e. zero variance / temperature). We perform a sweep over these hyperparameters, and thus our results implicitly include these models as potential special cases.
> 2. We have reworked these sentences to make them a bit clearer. Our aim was not to suggest that the Gumbel distribution over endpoints, itself, has a strong inductive bias toward modeling hierarchies, but rather point out that Vilnis et al. 2018 demonstrate box embeddings effectively model hierarchies and Dasgupta et al. 2020 demonstrate that using the Gumbel distribution over endpoints makes this even more effective.
> 3. We agree, and although GumbelBox was not the main focus of this paper (having been introduced in Dasgupta et al. 2020) we have updated the paper to include a short discussion of the Gumbel distribution.
> 4. This is true, however: (1) the meet operation serves primarily to justify the theoretical properties of the box as a lattice, (2) we do not directly train or evaluate using the meet operation, as it is not needed for our queries. That being said, for a well-trained model of hierarchies, the meet is still meaningful, as the meet of a node and one of it’s descendents would simply be the node itself, and the meet of any two arbitrary nodes will provide the smallest containing box which, itself, is contained in the closest common ancestor node.
> 5. Thank you for pointing this out, there is a typo here which we have corrected to “(a≤1b)∧(b≤2c)→(a≤2c)”.  We’ve also added “(a≤2b)∧(b≤1c)→(a≤2c)”, which corresponds to your example (Bird HasPart Wing, Wing IsA Appendage => Bird HasPart Appendage).
> 6. This approach can easily be extended to more than two hierarchical relations by learning additional transformations, however we are not aware of any dataset which contains three or more hierarchical relations in sufficient quantity/density such that modeling all three jointly would lead to improved inference. IsA and HavePart are both prevalent and fundamental relations, however, and it is our belief that modeling them correctly will lead to benefits on additional non-hierarchical relations, which is a major aim of our future work.
> 7. Note that the gumbel beta is a global parameter which is the same for all embeddings. (Dagupta et al. 2020 mention this is a requirement for GumbelBox to be closed under intersection.) In our experiments, we follow Dasgupta et al. 2020 and tune β on a validation set using Bayesian hyperparameter optimization. While it is possible to learn β via gradient-descent on the training set, it is likely that this would also quickly lead to local minima with very small β (due to the influence of negative samples), and thus it seems more appropriate for this to be treated as a global hyperparameter selected based on validation set performance, or even annealed throughout training.
> 8. We have tried more complicated transformations, including shallow MLPs, but a simple linear transformation actually outperforms them. This is likely due to a fundamental difference in the way that MLPs interpret their input (encoding information using the vector-space structure) and the way these vectors are used in the GumbelBox model, where they are eventually used to calculate ratios of expected intersection volumes.
> 9. One way to think about this is that the “IsA-Bird” box represents all things which are birds, and the “HasPart-Wing” box represents all things which have wings. A bird is something which has a wing, so it belongs in the “HasPart-Wing” box. A box for “HasPart-Bird”, on the other hand, would represent all things which have birds as a part of them, so perhaps the “IsA-Flock” would be inside this box (if such a Flock node existed). We have clarified this point in the updated version of the paper.

---

> > ### Author Response · Authors · 2020-11-18
> > **Clarification regarding Gumbel Box (cont.)**
> >
> > 10. Note that we discuss training in section 3.2 and prediction in section 5.3. In short,  any edge from X to Y (eg. Person to Man) can be modelled as P(X|Y) =1 (eg. P(person|man) =1, would ensure “person” box contains “man” box). In case of negatives, P(X|Y) = 0. We achieve this via gradient descent using KL-divergence loss between P(Box(X)| Box(Y)) and given P(X|Y). During inference, we predict an edge if P(Box(X)|Box(Y)) > threshold, the threshold used for this classification is determined by maximizing the F1 score on the validation set.
> > 11. Yes, the order embedding model reported here is the same as that described in Patel et al. 2020. In Vilnis et al. 2018, it was observed that box embeddings have strictly greater representational capacity than order embeddings, where (for example) the min coordinate of all boxes are fixed to the origin.
> > 12. In all of our experiments, the evaluation data is constructed with a ratio of positive and negative to be 1:10 as mentioned in section 5.3. (GumbelBox itself was evaluated on the dataset you mention in Dasgupta et al. 2020, section 5.3, where it is shown to significantly outperform the model in Li et al. 2019.)

---

> > > ### Author Response · Authors · 2020-11-24
> > > **Three Hierarchical Relations:**
> > >
> > > As requested, we have also created an evaluation involving 3 hierarchical relations:
> > > IsA (hypernym + instance_hypernym)
> > > PartOf (part_meronym + substance_meronym)
> > > member_meronym
> > > We include compositional edges between member_meronym and IsA in the same manner as for PartOf and IsA. Preliminary experiments on a representation task suggest a similar trend for model performance, where the box-to-box transformation model outperforms multi-relational hyperbolic baselines.
> > >
> > >
> > > | 3-Relation Hierarchy  |   ---------------------------------- --------------| F1 score   |
> > >
> > >     | MuRP |                                   |   0.17  |
> > >     | RotE   |                                   |   0.53  |
> > >     | AttH   |                                   |   0.53  |
> > >     |  RotH  |                                  |   0.54  |
> > >     | Box-to-box transform |                    | 0.71     |
> > >
> > >
> > > If accepted, we will include these results in the camera-ready, along with evaluations on the other baselines as well as generalization evaluations for this three-hierarchy setting.

---

### Official Review · AnonReviewer4 · 2020-10-24
**Seemingly incremental contributions, but with significant empirical improvements.**

**Rating:** 6
**Confidence:** 4

**Review:**

This paper builds upon the work of Patel et al. (2020) in modeling two hierarchies jointly within the box embedding framework. It also incorporates the GumbelBox formulation of Dasgupta et al. (2020) to resolve local identifiability issues during training.

The contribution of the paper seems to only lie in the learning of a function \phi that maps entity boxes to HasPart-* boxes. This function constrains the HasPart-* boxes in two ways: (a) their "minimum" corners remain at the same relative positions as their corresponding entity boxes, and (b) their lengths are scaled proportionately in each dimension. In contrast, Patel et al. (2020) does not have these constraints in their model. I find the novelty of these constraints to be incremental, especially in view that the joint hierarchy problem and evaluation methodology have already been formulated by Patel et al (2020) in the context of box embeddings. Though seemingly straightforward, the constraints do help the paper to improve upon the state of the art by significant margins in the experiments.

The paper is well organized and clearly written for the most part, but the exposition can be improved in some areas.

* Section 4.1, (a <_1 b) ^ (b <_2 c) => (a <_2 b): Could the authors provide examples of what <_1, <_2, a, b, and c represent?
I interpret "a <_1 b" to be b IsA a, "b <_2 c" to be c HasPart b, which then leads to "c HasPart a". This means that the consequent should (a <_2 c) rather than (a <_2 b), no?

* Section 1, 3rd para: hiearchy->hierarchy,

* Section 1, 4th para: dialate->dilate

* Section 5.3, 1st para, "for those edges in table 5" -> should be "table 2"?

---

> ### Author Response · Authors · 2020-11-18
> **Simple but effective**
>
> Thank you for recognizing our contribution and promising experimental results as shown in the paper, and for providing the specific corrections, we have updated the paper accordingly. Although the proposed method is simple, prior to this work it was unclear how to effectively enable sharing of parameters between boxes for the purpose of transforming one graph to another. Based on other reviewer’s requests, we have also run additional baselines, including recent hyperbolic embedding methods, and find that our relatively simple model significantly outperforms them. A further contribution of our paper is the additional analysis of the model’s ability to generalize. Patel et al. 2020 was purely a representation task, which was appropriate for the model structure proposed, however sharing parameters allow us to evaluate our model for generalization capability, and we include a thorough breakdown and analysis of various types of generalization this model is capable of performing.

---

### Official Review · AnonReviewer1 · 2020-10-28
**An interesting paper.**

**Rating:** 8
**Confidence:** 4

**Review:**

This paper deals with tree-like structure embedding with box embedding on the lattice (poset). This paper is well-motivated and well-presented. Though there is a limitation on data structure, this paper still presents a novel idea in this area. This method also achieved promising results in experiments. Thus, I would like to recommend to accept this paper.

---

> ### Author Response · Authors · 2020-11-18
> **Thank you!**
>
> Thanks R1 for recognizing our contribution in proposing this novel method and promising experimental results in representing tree-like structures!

---

### Decision · Program_Chairs · 2021-01-07
**Final Decision**

**Decision:**

Reject

**Comment:**

The paper is concerned with modeling multi-relational data with joint hierarchical structure. For this purpose, the authors extend box embeddings to multi-relational settings, supporting the modeling of cross-hierarchy edges and generalizing from a subset of the transitive reduction. The reviewers highlight that the paper is, overall, well-written and organized, relevant to the ICLR community, and that the proposed method offers promising experimental results. Furthermore, the author's rebuttal clarified some concerns of the initial reviews (e.g., relation to GumbelBox, comparison to additional baselines etc.) and improved the manuscript.

However, after rebuttal there exist still concerns regarding the current version. Reviewers raised concerns regarding novelty, clarity, and the empirical evaluation (importantly modeling more than 2 relations; it would also be good to understand more clearly why some of the newly added multi-relational hyperbolic baselines perform worse than uni-relational Poincare embeddings). While the paper and the proposed method clearly have promise, I agree with reviewers that the manuscript would require an additional revision to clarify these points. Given the positive aspects of the paper, I'd strongly encourage the authors to revise and resubmit their work given this feedback.